# Does putting down your smartphone make you happier? the effects of restricting digital media on well-being

**Lisa C. Walsh** *, Annie Regan, Karynna Okabe-Miyamoto, Sonja Lyubomirsky

Department of Psychology, University of California, Riverside, CA, United States of America

* lisawalsh08@gmail.com

## Abstract

Both scientists and laypeople have become increasingly concerned about smartphones, especially their associated digital media (e.g., email, news, gaming, and dating apps) and social media (e.g., Facebook, Instagram, Snapchat). Recent correlational research links substantial declines in Gen Z well-being to digital and social media use, yet other work suggests the effects are small and unnoteworthy. To help further disentangle correlation from causation, we conducted a preregistered 8-day experimental deprivation study with Gen Z individuals ($N = 338$). Participants were randomly assigned to one of four conditions: (1) restrict digital media (i.e., smartphone) use, (2) restrict social media use, (3) restrict water use (active control), or (4) restrict nothing (measurement-only control). Relative to controls, participants restricting digital media reported a variety of benefits, including higher life satisfaction, mindfulness, autonomy, competence, and self-esteem, and reduced loneliness and stress. In contrast, those assigned to restrict social media reported relatively few benefits (increased mindfulness) and even some potential costs (increased negative emotion).

**Data Availability Statement:** Data, measures, and R code are available on the Open Science Framework (OSF) and may be found at https://doi.org/10.17605/OSF.IO/VPEKX.

## Introduction

Smartphones have become ubiquitous in industrialized and developing nations alike. There are over 7 billion smartphone users worldwide, and that number is expected to reach 7.7 billion in the next five years [1]. Smartphone penetration rates have exceeded 50% in many countries, including Brazil (66.6%), Mexico (61.5%), China (68.4%), Iran (74.4%), France (82.6%), and the United States (81.6%) [2]. Moreover, people report spending ever increasing amounts of time on their smartphones—currently up to about 2.5 to 5 hours per day [3–8]. Clearly, smartphones have permeated every aspect of daily life for many people around the globe. These devices allow for nearly infinite possibilities, including listening to obscure bands on-demand, messaging friends on the go, finding romantic partners, getting driving directions, sharing photos on social media, and Zooming with co-workers abroad. Although these technologies are valuable, rewarding, and convenient in myriad ways, both scientists and laypeople have become increasingly concerned with their potential costs.

Narratives about the harmfulness of smartphones have seeped into the general lexicon of global society. The Netflix documentary, "The Social Dilemma," amplified public awareness of

**Funding:** The authors received no specific funding for this work.

**Competing interests:** The authors have declared that no competing interests exist.

this issue by emphasizing how social media connects, divides, controls, distracts, and manipulates people, as well as by showcasing dramatic enactments of teens unable to navigate a single family dinner without their devices [9]. A variety of self-help books have already gained popularity by promising strategies for "outsmarting" or "breaking free" from one's smartphone [10,11]. Popular news media outlets also highlight the darker aspects of smartphone use, ranging from fatalities that occurred while taking selfies to teens allegedly developing bone spurs on their skulls (i.e., "growing horns") [12,13]. Such coverage may lead laypeople to believe that smartphones are inherently detrimental. However, within the scientific community, this issue has been the subject of intense debate. Some studies suggest smartphones are harmful to well-being [14,15], while others argue the negative effects are, in fact, negligible [16,17].

Are smartphones making people unhappy? This is a challenging question to conclusively answer. Nearly two decades into the smartphone revolution, there is no way to put this technology back into the proverbial box from whence it came. However, we can attempt to tackle a more specific question: Can putting down your smartphone (at least some of the time) make you happier? In the present study, we conduct a short longitudinal (i.e., "shortitudinal") intervention to test whether restricting smartphone use improves well-being and related outcomes, such as self-esteem and loneliness [18]. Below we discuss the different types of smartphone media—digital and social—that past studies suggest may have detrimental effects.

## Digital media and well-being

Smartphones would not be that useful without their associated digital media, which has been defined as the "video, audio, software, or other content that is created, edited, stored, or accessed in digital form" [19]. Importantly, digital media can be accessed via computers, iPads, TVs, and even video game consoles. But with the rise of smartphones, digital media frequently come in the form of messaging, social media, email, music, gaming, video streaming, and other apps.

In correlational studies, Twenge and colleagues found that Gen Z (or iGen) adolescents in the U.S. ($Ns$ = 506,820 to 1.1 million) experienced decreases in psychological well-being and increases in depressive symptoms (including a rise in suicide rates) from 2010 onward [14,15]. These adolescents also reported drops in self-esteem and domain satisfaction. To account for these shifts, the researchers observed that adolescents who spent more time on screen activities (e.g., scrolling social media, texting, browsing the Internet, gaming) and less time on non-screen activities (e.g., socializing, exercising, doing homework) reported lower well-being and higher depression. Adolescents who spent a small amount of time on screens were happiest. Partial correlations between happiness and screen activities (including demographic controls) ranged from $r$ = –0.01 (reading news online) to $r$ = –0.11 (Internet use).

Notably, the observed rise in depressive symptoms and decline in well-being closely followed the rise of smartphone technology. The first iPhone was introduced in 2007 [20], and among U.S. adolescents, smartphone ownership leapt from 37% in 2012 to 73% in 2015 [21]. The above correlational investigations suggest that digital media—specifically, time spent on smartphone-related screen activities—may be harmful to well-being.

However, drawing on additional correlational studies, Orben and Przybylski argued that digital media use may have small effects on well-being and related constructs (e.g., depression, self-esteem) [16,17]. In one study, the researchers analyzed three large-scale datasets (total $N$ = 355,358) and found a negative but small association ($\beta$ = –0.042) between digital media use and well-being. Comparing this association to other activities, they concluded that digital media's negative effect on well-being was comparable to that of eating potatoes ($\beta$ = –0.042) and wearing eyeglasses ($\beta$ = –0.061). Another follow-up investigation by Orben and Przybylski

analyzed both self-report and time-use diary measures and again concluded that digital media use is not meaningfully linked to Gen Z well-being, depression, or self-esteem. In aggregate, the median association between self-reported screen time and well-being was $\beta = -0.08$, but time-use diary measures reduced this estimate to close to zero ($\beta = -0.02$).

Due to the difficulty in inferring causality from correlational research, a few experimental studies have begun to test the effects of restricting smartphone use, at least during short time periods or specific circumstances. For example, in two experiments, college students were directed to find a campus library with or without their smartphone. Relative to those using their smartphones, students not using them arrived at the building feeling more socially connected, but it took them longer to find the building and the difficulty of the task appeared to make them less happy [22]. In another investigation, groups of three to five friends or family members out to dinner at a local café were directed to keep their phones on them or put them on silent and set them in a locked, closed container on the table. The diners who kept their phones reported more distraction, as well as lower interest, enjoyment, and well-being during dinner [23].

Other studies have tested the effects of limiting smartphone use in specific, targeted ways. In one study, participants maximized phone interruptions for 1 week by keeping push notification alerts on and their phones within their reach or sight [24]. The next week, participants minimized phone interruptions by keeping alerts off and their phones away. Participants reported higher levels of inattention and hyperactivity when alerts were on than when alerts were off. Higher levels of inattention, in turn, predicted lower productivity and well-being. Another experiment sought to determine whether batching smartphone notifications might improve happiness [25]. Relative to receiving notifications as usual, hourly, or no notifications at all, batching smartphone notifications 3 times per day increased well-being.

Additionally, several experiments have examined the effects (e.g., on anxiety, attention) of constraining smartphone use for up to several hours in the lab while completing tasks, such as reviewing class materials or solving puzzles [26–29]. A few studies have explored general smartphone restriction outside the lab. In one study, participants assigned to an experimental group who abstained from smartphone use for 72 hours did not exhibit notable changes in positive or negative emotions, relative to the control group who used their smartphones as usual [30]. In another daily diary study, 19 participants were directed to abstain from smartphone use for 10 days. At the end of the study, participants reported reduction in their smartphone cravings and feelings of exhaustion [31,32].

The most comprehensive general smartphone restriction study to date involved a 7-day intervention, in which German participants were randomly assigned to one of three conditions that directed them to (1) abstain from smartphone use, (2) reduce smartphone by an hour per day, or (3) use their smartphones as usual [33]. The results showed that, relative to control, both reducing and abstaining from smartphone use improved participants' life satisfaction, depression, and anxiety. These effects were stronger and more stable in the reduction group than the abstinence group at a 4-month post-intervention follow-up. In other words, complete abstinence was not necessary to improve well-being, and there may even be a potential "sweet spot" for smartphone use. Notably, this study left two gaps in the literature that the present study addresses. First, participants self-reported how much time they spent on their smartphones, and such self-reports are notoriously unreliable [8]. Second, the researchers did not assess several other important well-being-related outcomes (e.g., positive emotions, loneliness).

In sum, correlational research on digital media has yielded two competing messages. Some psychological scientists conclude that digital media use may harm well-being, while others infer that it has no meaningful impact. However, disagreements appear to be less about

correlational effect sizes (which are often highly similar) than about how to interpret them. The experimental research so far has found that limiting digital media use on smartphones in targeted ways (e.g., at dinner, batching notifications) often bolsters well-being, but limiting it sometimes backfires (e.g., when trying to find an unfamiliar location). Although one important previous study restricted general digital media (i.e., smartphone) use in daily life for about a week [33], additional smartphone restriction experiments may serve several useful functions, including replicating existing effects, providing more data for future meta-analyses, collecting objective (rather than self-reported) indicators of smartphone use, and testing additional outcomes (e.g., positive emotions, social connectedness). Such experiments may help further disentangle correlation from causation, and better elucidate the strength and direction of effects between digital media and well-being.

## Social media and well-being

Broadly defined, social media is a type of digital media that allows individuals to create and share user-generated content [34], such as blog posts, tweets, and videos. The most common types of social media are often referred to as social networking sites [35], and include examples like Facebook, Instagram, TikTok, Snapchat, and Twitter (recently renamed "X"). Although these sites can be accessed via computers and tablets, the most popular social networking sites are predominantly used on smartphones [36]. Like smartphones, social media use has become pervasive. As of 2021, approximately 70% of U.S. adults reported using some form of social media [37]. At the current rate of growth, it is projected that nearly 6 billion people will be using social media worldwide in 2027 [38].

Notably, emerging evidence suggests that social media may be an especially harmful component of digital media, exerting adverse effects on well-being. Several studies have specifically focused on prompting users to reduce the amount of time they spend on Facebook. One of the most frequently cited studies ("The Facebook Experiment") recruited 1,095 Danish individuals, and randomly assigned them to stop using Facebook for a week or keep using it as usual [39]. Participants who gave up Facebook experienced increases in life satisfaction, and their emotions became more positive. Effects were strongest for users who initially used Facebook heavily, reported feeling high Facebook envy, and typically used Facebook passively (i.e., scrolling their news feeds).

A longer Facebook deprivation study found similar effects. The researchers measured 2,743 Facebook users' willingness to deactivate their accounts for 4 weeks, then paid a randomly selected subset to do so [40]. At posttest, the users who deactivated their accounts reported increases in positive emotions, subjective happiness, and life satisfaction, relative to those who did not deactivate their accounts. In another study (a natural experiment), an Israeli company banned employees from using Facebook altogether at the office, then later differentially restricted its use [41]. Employees who continued to use Facebook engaged in more social comparison and showed diminished happiness. However, these effects only applied to the younger half of the sample, and only if those young people believed others had more positive experiences than they did.

Other studies have attempted to assess whether using social media in specific ways may produce different well-being outcomes. One study brought participants into the lab and directed them either to use Facebook passively (e.g., by scrolling through their newsfeed and looking at friends' pages) or actively (e.g., by posting status updates or directly messaging friends) [42]. Neither group demonstrated changes immediately following the manipulation, but participants in the passive use group showed a significant drop in affective well-being at the end of the day. However, a recent scoping review of 40 studies concluded most studies did not

support the notion that active social media uses increases well-being, whereas passive social media use decreases well-being [43].

The studies reviewed above suggest that restricting Facebook provides well-being benefits. What about other social media services? As of 2024, Facebook was still the top social network in the U.S., with 3.1B unique monthly visitors, but numerous other social networking sites are extremely popular, such as YouTube (2.5B monthly users), Instagram (2B monthly users), TikTok (1.6B users), Snapchat (800M monthly users), and X/Twitter (611M monthly users), among others [44].

Limiting Facebook alone likely does not restrict all (or even most) social media use, because participants directed to restrict Facebook may just begin to use another social media app instead. Moreover, Facebook is no longer the most popular social media platform among Gen Z individuals [45]—the age group about whom much of the concern about screen time and mental health has focused. Teens are abandoning Facebook at an ever-increasing rate in favor of alternative social media, such as Instagram, Snapchat, and TikTok [46]. Thus, examining other social media platforms is a compelling next step for the field.

Recently, a few studies directing participants to restrict other types of social media have emerged. One study assigned undergraduates to either limit Facebook, Instagram, and Snapchat use to 10 minutes per platform per day, or to use social media as usual for 3 weeks [47]. Relative to controls, the participants assigned to limit their social media use showed significant reductions in loneliness and depression. However, no significant differences emerged between the two groups for social support, anxiety, self-esteem, or autonomy. Notably, this study left several other social media services unrestricted (e.g., Twitter, Reddit), which participants could have turned to instead. Also, some of the most used well-being measures in the psychological literature (e.g., positive emotions, life satisfaction) were not assessed. In another article with three experiments (total $N = 600$), participants were assigned to one of two conditions: a normal-use social media day or an abstinence day [48]. Taking a short 1-day break from social media did not significantly improve positive affect, negative affect, or self-esteem; and appeared to display some backfiring effects—harming feelings of social relatedness (a type of need satisfaction) and satisfaction with one's day. These findings are compelling, but only apply to a single day.

Taken together, previous research indicates that fully restricting specific social networking sites (e.g., just restrict Facebook) or in specific ways (e.g., use social media actively vs. passively) may yield psychological benefits (e.g., increased well-being, reduced loneliness). Notably, most experimental studies deploy a "use social media as usual" vs. "stop using social media" approach. However, participants instructed to use social media as usual may not represent a strong control group that accounts for experimental demand effects. The present study sought to build on previous findings by restricting all social media in daily life while employing an alternative activity control condition. Using this approach, we assessed subsequent effects on well-being and related psychosocial constructs. For a comprehensive overview of digital and social media research, we provide a detailed list of correlational and experimental studies along with their relevant outcomes and key findings in the (see S1 Table in S1 File).

## The present study

In the present study, we sought to further explore the effects of digital media and social media on well-being by experimentally manipulating it in daily life. Instead of asking participants to limit their media use for a short period of time (e.g., at dinner, for one day) or in specific, targeted ways (e.g., batching smartphone notifications), we asked them to actively restrict their digital media and social media use for about 8–10 days.

Our method differs from past work in a few important ways. First, we collected objective indicators of smartphone and social media use, instead of just relying on self-reports. Second, we tested effects on several important outcomes (e.g., positive emotions, social connectedness, mindful attention) not addressed in previous studies. Third, we deployed both an active and measurement-only control as a way of better testing for controlling for demand effects. Finally, we included two experimental conditions separately testing the effects of restricting smartphone and social media use in the same study, which allowed us to examine whether one type of restriction provided greater benefits.

Towards these ends, we recruited Gen Z individuals, and randomly assigned them to one of four conditions in a between-subjects design: (1) restrict digital media use (Digital Diet), (2) restrict social media use (Social Diet), (3) restrict water use as an active control (Water Diet), or (4) restrict nothing as a measurement-only control (No Diet). Will restricting digital media and social media use as much as possible for about a week improve psychological well-being and related constructs?

## Hypotheses

We tested the preregistered hypotheses listed below. Overall, we hypothesized that restricting digital media and social media would have psychological benefits. However, given the debate in the correlational literature mentioned above, we anticipated the possibility of finding null or even backfiring effects, as such effects would be valuable and informative to document.

**Hypothesis 1.** Relative to our two control groups (Water Diet, No Diet), participants assigned to restrict their smartphone digital media use (e.g., gaming, social media, entertainment, online news apps; Digital Diet) will demonstrate greater increases in positive affect, happiness, life satisfaction, mindful attention, self-esteem, self-reported health, connectedness, autonomy, and competence, as well as larger decreases in negative affect, depression, stress, and loneliness.

**Hypothesis 2** Relative to controls (Water Diet, No Diet), participants assigned to restrict their social media use (e.g., Facebook, Instagram, Twitter, Snapchat; Social Diet) will demonstrate greater increases in positive affect, happiness, life satisfaction, mindful attention, self-esteem, self-reported health, connectedness, autonomy, and competence, as well as larger decreases in negative affect, depression, stress, and loneliness.

**Exploratory.**   Because we did not have specific a priori hypotheses about which condition might outperform the other, we preregistered our comparisons of the Digital Diet vs. Social Diet as exploratory.

## Method

### Transparency and openness

We report our target sample size, as well as all data exclusions, manipulations, and measures below. We preregistered our hypotheses on the Open Science Framework (OSF): https://osf.io/bv9dj. Data, materials, and R code (using version 4.0.2) are also available at: https://osf.io/vpekx/.

### Participants

We recruited undergraduate students from the psychology department's online participant pool at a large public university. The study required the following eligibility criteria: Participants had to be at least 18 years old, read and write English fluently, own an iPhone running

iOS 12 or later with *Screen Time*, and use social media at least four to six times per week. Students received course credit as compensation for their participation. Those who completed the entire study and reported putting at least minimal effort toward their assigned activity instructions received an extra $10 Amazon digital gift card bonus. The study was approved by the UCR Institutional Review Board–Socio-Behavioral (IRB-SB) #HS-12-112. Written consent was obtained.

We aimed to recruit 100 participants per condition (target $N = 400$) [49]. A total of 414 participants completed at least one survey (Time 1/pretest). To help ensure the credibility of responses, we also preregistered several exclusion criteria. Specifically, participants were excluded from analyses if they answered 15 simultaneous questions with the same response (i.e., "straightlining"; 3 excluded), reported they did not restrict their digital media, social media, or water use at all (6 excluded), put no effort into their assigned activity (5 excluded), and/or their daily average Time 2 media use exceeded their Time 1 use (33 excluded in the Digital Diet and Social Diet conditions only). A CONSORT flow diagram for the study is presented in Fig 1.

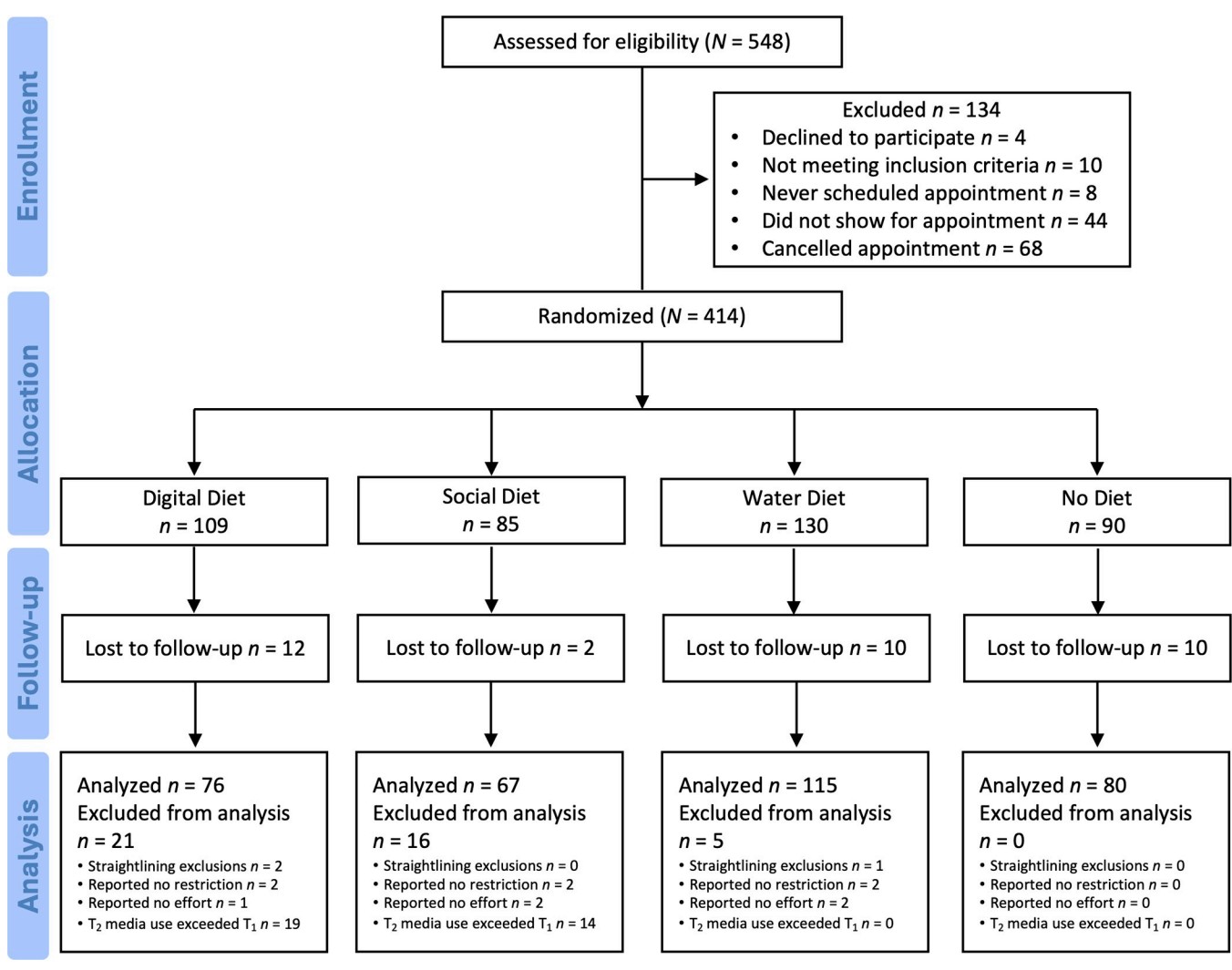

**Fig 1. CONSORT flow diagram.**

We also originally planned to exclude participants who answered "No" to a Self-Reported Single Item (SRSI) question: "In your honest opinion, should we use your data in our analyses in this study?" [50]. Surprisingly, a sizable subset of participants (32) answered "No" to this question—more than in previous studies—and given its ambiguous interpretation, we decided to discard the SRSI question exclusion criteria and keep those participants in the analyses. Finally, 34 participants were lost to follow-up and did not complete the Time 2/posttest survey. Please note that some participants were excluded because they matched multiple exclusion criteria (e.g., straightlining and not restricting at all), yielding a final sample of $N = 338$. Overall, we followed all our preregistered data exclusions except the proposed SRSI question exclusion. A sensitivity power analysis using G*Power indicated that the smallest effect we could detect with 80% power and a one-tailed test was $r = .17$ [51].

Among the 338 participants ($M_{age} = 19.4$, $SD = 2.4$), almost all (97.9%) were born in 1995 or later, placing them in the generation known as Gen Z (or iGen), the first generation to enter adolescence with smartphones [52]. We chose to sample Gen Z individuals because they tend to experience high rates of social isolation, loneliness, fear of missing out, and poor mental health outcomes—and have been the focus of much of the correlational research described earlier [14–17]. The participants were also predominantly female (78.1% female) and single (64.2%). They came from a variety of ethnicities, including Asian (40.5%), Hispanic (34.9%), White (10.7%), Black (3.8%), other (3.9%), and more than one (6.2%). Participants also reported a range of household incomes: 26.4% reported that their families earned less than $30,000 a year; 20.1% earned between $30,001 and $60,000; 19.8% earned between $60,0001 and $100,000; 20.7% earned over $100,000; and 13% did not know their household income. Many student participants also worked part-time (31.4%).

## Procedure

Fig 2 presents an overview of the study timeline. To reduce demand effects, participants were ostensibly recruited for a "Daily Habits Study"—a study examining daily habits (e.g., drinking, eating, exercising), behaviors (e.g., reading, watching TV, smartphone use, water use), thoughts, emotions, and physical health. The duration of the study averaged 8 days (range = 7–-13 days) with two time points. At both Time 1 ($T_1$) pretest and Time 2 ($T_2$) posttest, participants visited our lab in-person. Data collection began in February 19, 2019 and ended on March 2, 2020, just prior to the university's transition to online learning due to the COVID-19 pandemic.

**Time 1 ($T_1$ / Pretest).** To begin the study, participants signed a consent form. Then research assistants (RAs) collected dried blood spots (DBS; 3–5 drops of blood) via finger prick for collection on protein saver cards for later laboratory analysis of leukocyte gene expression. The DBS analyses are beyond the scope of this investigation and are not presented here. After DBS collection, participants were directed to a private computer to complete an online survey of outcome measures and demographic information.

At the end of the $T_1$ survey, participants were randomly assigned to one of four conditions (Digital Diet, Social Diet, Water Diet, or No Diet) that varied with respect to their daily activity instructions. Participants were assigned to condition using a Qualtrics randomizer block in the survey flow, with the "Evenly Present Elements" option deselected to maintain genuinely random assignment. In other words, there were no stratification factors used to balance across groups. This method, however, led to the creation of groups that were somewhat unequal in size. See Supplemental Materials for condition instructions.

Participants in the Digital Diet condition ($n = 76$) were instructed to limit their digital media use on their smartphones. They were allowed to use their smartphones for practical

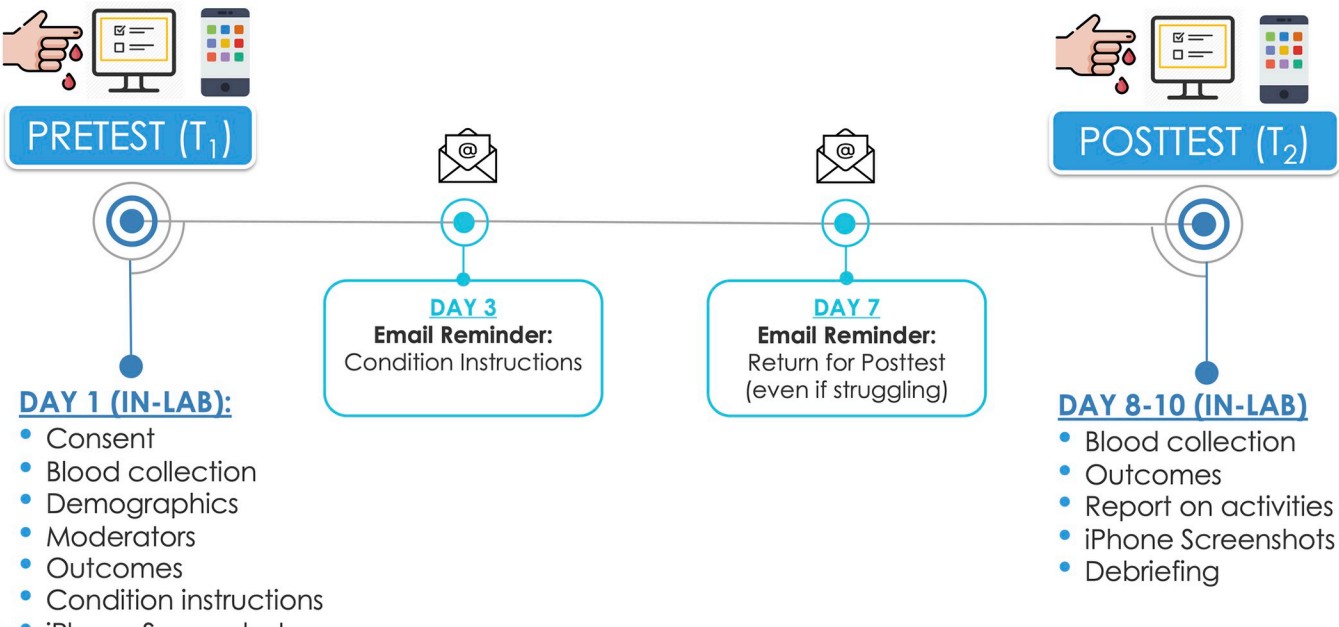

**Fig 2. Study timeline.** *Note*. Study timeline with $T_1$ (pretest) on Day 1 and $T_2$ (posttest) on Day 7–13. This figure has been designed using images from Flaticon.com.

purposes (e.g., to obtain GPS directions, to answer work emails), but were instructed to restrict their use as much as possible and to stop using any non-necessary apps (e.g., Facebook, Tetris, Hulu, CNN).

Participants in the Social Diet condition (*n* = 67) were directed to stop using social media during the intervention period. We provided them with recommendations about how to accomplish this aim (e.g., set a Screen Time Social Networking app limit, delete social media apps off their iPhones), as well as a list of social media apps/sites (e.g., Facebook, Instagram, Twitter, Snapchat) to avoid.

We also included two control conditions. To confirm that restriction alone (e.g., putting effort into doing something worthwhile and feeling good about it) was not driving effects, our first control condition was a Water Diet (active control) group (*n* = 115), in which participants were directed to restrict their water use. That is, they were asked to use less water when they washed their hands, brushed their teeth, took showers, washed dishes, etc., but not to restrict how much water they drink.

Our second control condition was a No Diet (measurement-only control) group (*n* = 80), in which participants only completed measures. They did not receive any instructions regarding their digital media, social media, or water use. The goal of this control condition (as well as the Water Diet group) was to include a subset of participants who continued using digital and social media as usual without prompting them to monitor and/or change their behavior. We made every effort to reduce the salience of tracking digital media use for these groups, as we were concerned that monitoring it may change it. For example, a systematic review of fitness tracking technologies (e.g., Fitbit, Nike+) found that self-tracking can prompt individuals to increase their physical activity levels [53].

Attrition was fairly comparable across conditions, with the Social Diet group demonstrating the lowest attrition rate (2.3%) and the No Diet condition demonstrating the highest (11.1%).

After students finished the survey and received their condition instructions, RAs helped them take screenshots of the *Screen Time* section of their iPhone in their Settings app. Screen

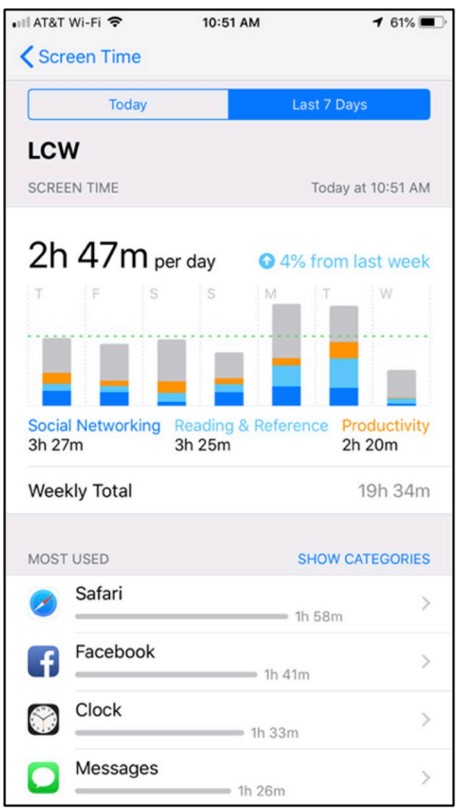
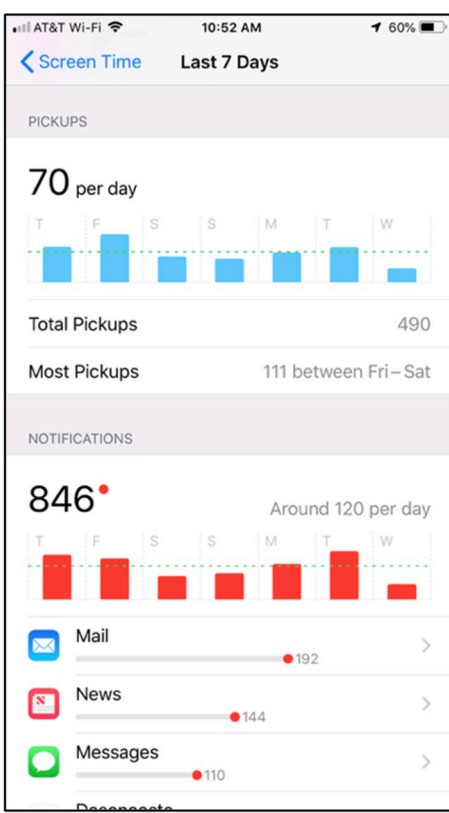

**Fig 3. Example iPhone screen time screenshots.** *Note*. Example screenshots were collected using an Apple iPhone 7 Plus running mobile operating system iOS 12.

Time is a feature of Apple's mobile operating system (iOS 12 and later) that provides various iPhone user characteristics, such as the average amount of time users spend on their iPhone including time spent on specific apps, as well as the number of times users picked up their phones and received push notifications (see Fig 3). Once captured, iPhone Screen Time screenshots were emailed to a general study email for later transcription, coding, and analysis.

**Time 2 ($T_2$ / Posttest).** At $T_2$, participants returned to the lab for a visit that was similar to the $T_1$ visit described above. Participants first provided another DBS sample, then completed a second posttest survey of outcomes. The survey asked them about their experiences during the past week and provided a debriefing statement. Finally, RAs collected a second set of Screen Time screenshots from participants' iPhones.

## Measures

Participants completed the following measures at $T_1$ and $T_2$, rating each measure over the "past week (last 7 days)."

**Brief happiness and satisfaction.** Adapted from Monitoring the Future (MtF), a large, multi-decade longitudinal survey of U.S. adolescents, we measured recent happiness and satisfaction with two, brief single items [15,54,55]. To measure happiness, participants were asked, "Taking all things together, how would you say things are these days—would you say you're very happy, pretty happy, or not too happy these days?" (1 = *not too happy*; 3 = *very happy*). To measure satisfaction, participants were asked, "How satisfied are you with your life as a whole these days?" (1 = *completely dissatisfied*; 7 = *completely satisfied*).

**Positive and negative emotions.** Affective well-being was assessed using a modified version of the Affect-Adjective Scale [56,57]). This 12-item measure taps a range of low and high arousal positive emotions (e.g., enjoyment/fun, relaxed/calm) and negative emotions (worried/anxious, dull/bored). Participants rated the extent to which they experienced each emotion in the past week on a 7-point Likert scale (1 = *not at all*, 7 = *extremely*). Scale reliabilities (McDonald's omegas [ωs]) ranged from .89 to .91 for positive affect and .75 to .82 for negative affect across timepoints.

**Life satisfaction.** We used the Satisfaction With Life Scale (SWLS) to assess participants' current satisfaction with their life in general [58]. The SWLS consists of five items (e.g., "In most ways my life is close to my ideal," "I am satisfied with my life"), which are rated on 7-point Likert-type scales (1 = *strongly disagree*, 7 = *strongly agree*). SWLS reliabilities ranged from ω = .85 to .88 across timepoints.

**Mindful attention.** Mindfulness (i.e., the extent to which participants are mindfully attending to the present moment) was measured with a 5-item short form of the Mindful Attention Awareness Scale (MAAS-Short) [24,59]. Example items include, "I found it difficult to stay focused on what was happening in the present" and "I found myself doing things without paying attention" (both reverse coded). Participants rated how they felt on a 6-point Likert scale (1 = *almost never*, 6 = *almost always*). MAAS-Short reliabilities ranged from ω = .80 to .85 across timepoints.

**Need satisfaction: Autonomy, competence, and connectedness.** We assessed three types of need satisfaction (feelings of autonomy, competence, and connectedness [or relatedness]) with a shortened 9-item version of the Balanced Measure of Psychological Needs (BMPN) [60]. This questionnaire includes 3 items each to assess autonomy (e.g., "I felt free to do things my own way"), competence (e.g., "I felt very capable in what I did"), and connectedness (e.g., "I felt a sense of contact with people who care for me"). Participants rated their level of agreement with each item on a 5-point Likert-type scale (1 = *not at all*, 5 = *much agreement*). Across timepoints, the reliabilities were ω = .74 to .80 for autonomy, ω = .76 to .77 for competence, and ω = .88 to .89 for connectedness.

**Depressive symptoms.** Depressive symptoms were measured with 6 items (e.g., "Life often seems meaningless," "I feel that I can't do anything right") from the Bentler Inventory of Depression [61]. Response choices ranged from 1 (*disagree*) to 5 (*agree*). Scale reliabilities for depression were ω = .87 to .90 across timepoints.

**Loneliness.** To measure loneliness, we administered a 6-item scale from MtF [55]. Participants indicated their level agreement on a 5-point Likert-type scale (1 = *disagree*; 5 = *agree*). Example items include "A lot of times I feel lonely" and "I usually have a few friends around that I can get together with" (reverse scored). Scale reliabilities for loneliness ranged from ω = .61 to .73 across timepoints.

**Self-esteem.** We also used the 6-item Rosenberg Self-Esteem scale [62]. Participants indicated their level of agreement (1 = *strongly disagree*; 5 = *strongly agree*) on items such as "Sometimes I think that I am no good at all" (reverse scored) and "I am able to do things as well as most other people." Reliabilities for self-esteem ranged from ω = .84 to .87 across timepoints.

**Stress.** Participants' stress levels were assessed with a 4-item short form of the 14-item Perceived Stress Scale [63]. Example items include, "How often have you felt you were unable to control the important things in your life?" and "How often have you felt things were going your way?" (reverse coded) Participants were asked to indicate how often they felt a certain way on a 5-point Likert-type scale (1 = *never*, 5 = *very often*). Reliabilities for stress ranged from ω = .74 to .76 across timepoints.

**Self-reported health.** Participants were also asked to report on their health-related quality of life using an adapted 5-item version of the SF-36 Health Survey [64]. Example items include

"Overall, how would you rate your health during the past week?" (1 = *very poor*, 6 = *excellent*) and "How much bodily pain have you had during the past week?" (1 = *none*; 6 = *very severe*; reverse coded). Because the SF-36 uses different scale points (e.g., 5-point and 6-point), each item was recoded on a value of 0 to 100 to create composites. Scale reliabilities for health were $\omega$ = .76 at both timepoints.

**Self-reported digital media time and social media time.** Prior to collecting objective time indicators (see below), we asked participants to estimate how much time they spent using digital media (i.e., smartphone time) in hours and minutes. To assess self-reported digital media (i.e., smartphone) time, participants were asked, "As accurately as possible, please estimate the total amount of time you spend using your smartphone on average per day." They were also required to provide a self-reported estimate of how much time they spent using social media. Specifically, we asked, "As accurately as possible, please estimate the total amount of time you spend using social media apps/sites on average per day. Please include time spent on all types of social media (e.g., Facebook, Instagram, Twitter, Snapchat) on all types of devices (e.g., iPhones, iPads, computers)."

**Objective digital media time and social media time.** To objectively assess how much time participants spent using digital media and social media on their iPhones over the past 7 days, at both the pretest and posttest visits, RAs helped participants capture screenshots of the *Screen Time* section of the Settings app on participants' iPhones (see Fig 2). RAs transcribed, coded, checked, and double-checked these Screen Time usage metrics on a shared Google Sheet, which we appended to survey data for analyses. Because the collected screenshots came from two different operating systems (iOS 12 and iOS 13) with various time durations (e.g., 3 to 13 days), we used Screen Time's "Weekly Total" estimate to create a daily average composite as an objective indicator of smartphone time. Additionally, we summed total time spent on various social media apps (e.g., Facebook, Snapchat, Twitter, Instagram), and created a daily average composite to assess objective social media time. We did not use iPhone's "Social Networking" app category because it includes time spent on apps generally not classified as social media (e.g., Messages, Phone, FaceTime). Notably, this measure only assessed objective social media time on participants' iPhones, which does not include time spent on other devices (e.g., computers, tablets). However, as mentioned earlier, a majority of people accessing social media services do so from their smartphones.

## Preregistered analytic plan

We preregistered our analytic plan on the OSF. To test Hypothesis 1, we subset the data to exclude the Social Diet group, then dummy coded condition to compare: (1) Digital Diet vs. Water Diet, (2) Digital Diet vs. No Diet, and (3) Digital Diet vs. Both Controls—with the Water Diet and No Diet control conditions coded as the reference group. To test Hypothesis 2, we followed a similar process, but this time omitted the Digital Diet condition to compare: (1) Social Diet vs. Water Diet, (2) Social Diet vs. No Diet, and (3) Social Diet vs. Both controls. Finally, we also compared the Digital Diet vs. Social Diet in an exploratory manner.

We used two preregistered statistical techniques to test our hypotheses: (1) Regressed change: Condition dummy codes predicting $T_2$ scores, controlling for $T_1$ scores, and (2) second-order latent growth models (SOLGM): Condition dummy codes predicting growth (i.e., slope) extracted from second-order latent growth models. Because both the regressed change and SOLGM statistical techniques produced highly similar (and often nearly identical) results, we focus solely on the regressed change analyses below. See (S1 Fig and S5 and S6 Tables in S1 File) for the SOLGM results.

In our regressed change models, regression coefficients were converted to partial correlations for ease of interpretation and comparability between models. Following past studies, the self-report and objective digital media and social media time use variables that were right-skewed and kurtotic were log-transformed before running analyses [7,65].

## Results

See Supplemental Materials for means and standard deviations by condition (S2 Table in S1 File) and bivariate correlations (S3 Table in S1 File).

### Manipulation checks

Did our participants change their behavior as directed? We first wanted to determine whether participants assigned to the Digital Diet and Social Diet conditions restricted their digital and social media use accordingly. Fig 4 shows pre-post difference scores by condition for both self-report and objective time use variables. Participants in the Digital Diet group showed greater decreases in both self-reported digital media time (partial $r = -.51$, $p < .001$; an average of –113 minutes/day) and objective digital media time (partial $r = -.57$, $p < .001$; an average of –115 minutes/day), relative to both controls.

The Social Diet participants also successfully reduced their social media use. Participants in the Social Diet group showed greater decreases in both self-reported social media time (partial $r = -.62$, $p < .001$; an average of –152 minutes/day) and objective social media time (partial $r = -.60$, $p < .001$; an average of –68 minutes/day), relative to both controls.

See Supplemental Materials for manipulation check individual condition comparisons (S4 Table in S1 File). Notably, the other individual condition comparisons (Digital Diet vs. Water

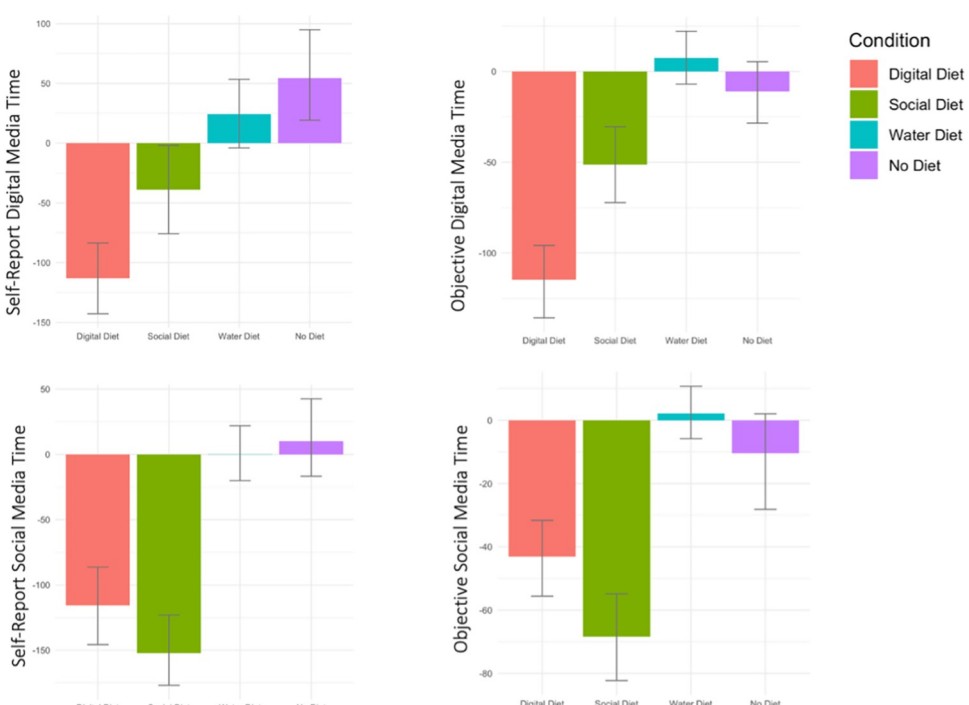

**Fig 4. Self-report and objective time difference scores by condition.** *Note.* $T_2 - T_1$ difference scores for self-report and objective digital media time and social media time (in minutes). For ease of interpretation, difference scores are presented in untransformed form.

Diet, Digital Diet vs. No Diet, Social Diet vs. Water Diet, Social Diet vs. No Diet) were also statistically significant at mostly $ps < .001$.

## Hypothesis 1. The effects of restricting digital media

According to our regressed change models (see Table 1), restricting digital media appeared to improve a variety of psychological outcomes.

**Digital diet vs. Water diet.**   Relative to the Water Diet control, participants in the Digital Diet group reported greater increases in life satisfaction (partial $r = .25$, $p < .001$), mindful attention (partial $r = .26$, $p < .001$), autonomy (partial $r = .21$, $p = .004$), competence (partial $r = .18$, $p = .015$), and self-esteem (partial $r = .30$, $p < .001$), as well as greater decreases in loneliness (partial $r = −.19$, $p = .009$) and stress (partial $r = −.21$, $p = .003$). We did not find statistically significant differences between the Digital Diet group and both control conditions for brief happiness, brief satisfaction, positive emotions, negative emotions, connectedness, depression, or health.

**Digital diet vs. No diet.**   Relative to the No Diet control, participants in the Digital Diet group reported greater increases in life satisfaction (partial $r = .21$, $p = .008$), mindful attention (partial $r = .20$, $p = .011$), competence (partial $r = .16$, $p = .046$), and self-esteem (partial $r = .24$, $p = .003$). We did not find statistically significant differences between the Digital Diet group and the No Diet control condition for brief happiness, brief satisfaction, positive emotions, negative emotions, autonomy, connectedness, depression, loneliness, stress, or health.

**Digital diet vs. Both controls.**   Relative to both controls, participants in the Digital Diet group reported greater increases in life satisfaction (partial $r = .21$, $p < .001$), mindful attention (partial $r = .21$, $p < .001$), autonomy (partial $r = .17$, $p = .006$), competence (partial $r = .15$, $p = .011$), and self-esteem (partial $r = .25$, $p < .001$), as well as greater decreases in loneliness (partial $r = −.13$, $p = .03$) and stress (partial $r = −.17$, $p = .006$). We did not find statistically significant differences between the Digital Diet group and both control conditions for brief happiness, brief satisfaction, positive emotions, negative emotions, connectedness, depression, or health.

## Hypothesis 2: The effects of restricting Social media

By contrast, restricting social media (Hypothesis 2) appeared to provide limited benefits, and even a few costs (see Table 2).

**Social diet vs. Water diet.**   Relative to the Water Diet control, the Social Diet group reported only significantly improved mindful attention (partial $r = .21$, $p = .005$). We found no significant differences on brief happiness, brief satisfaction, positive emotions, negative emotions, life satisfaction, autonomy, competence, connectedness, depression, loneliness, self-esteem, stress, and health.

**Social diet vs. No diet.**   Relative to the No Diet control, the Social Diet group experienced significant decreases in positive emotions (partial $r = -.17$, $p = .043$) and increases in negative emotions (partial $r = .23$, $p = .006$)—suggesting backfiring effects. In other words, participants in the Social Diet group had lower levels of happiness, joy, and serenity, as well as higher levels of sadness, anger, and boredom. We found no significant differences for brief happiness, brief satisfaction, life satisfaction, mindful attention, autonomy, competence, connectedness, depression, loneliness, self-esteem, stress, and health.

**Social diet vs. Both controls.**   Relative to both controls, the Social Diet group once again only experienced significant improvements in mindful attention (partial $r = .16$, $p = .012$). We found no significant differences on brief happiness, brief satisfaction, positive emotions, negative emotions, life satisfaction, autonomy, competence, connectedness, depression, loneliness, self-esteem, stress, and health.

**Table 1. Hypothesis 1 digital diet comparisons.**

| Variable | b | b SE | Partial r | Partial r 95% CI | | p |
|---|---|---|---|---|---|---|
| | | | | LL | UL | |
| **Hypothesis 1. Digital Diet vs. Water Diet:** | | | | | | |
| Brief Happiness | 0.09 | 0.07 | 0.09 | -0.05 | 0.23 | 0.215 |
| Brief Satisfaction | 0.19 | 0.16 | 0.09 | -0.06 | 0.23 | 0.227 |
| Positive Emotions | 0.06 | 0.13 | 0.03 | -0.11 | 0.18 | 0.639 |
| Negative Emotions | -0.18 | 0.11 | -0.12 | -0.25 | 0.03 | 0.113 |
| Life Satisfaction | 0.36 | 0.10 | 0.25 | 0.12 | 0.38 | < .001 |
| Mindful Attention | 0.41 | 0.11 | 0.26 | 0.13 | 0.39 | < .001 |
| Autonomy | 0.27 | 0.09 | 0.21 | 0.07 | 0.34 | 0.004 |
| Competence | 0.22 | 0.09 | 0.18 | 0.03 | 0.31 | 0.015 |
| Connectedness | 0.09 | 0.12 | 0.06 | -0.09 | 0.20 | 0.447 |
| Depression | -0.10 | 0.07 | -0.09 | -0.23 | 0.05 | 0.194 |
| Loneliness | -0.19 | 0.07 | -0.19 | -0.32 | -0.05 | 0.009 |
| Self-Esteem | 0.29 | 0.07 | 0.31 | 0.18 | 0.42 | < .001 |
| Stress | -0.23 | 0.08 | -0.21 | -0.34 | -0.08 | 0.003 |
| Health | 3.77 | 2.01 | 0.14 | -0.01 | 0.27 | 0.062 |
| **Hypothesis 1. Digital Diet vs. No Diet:** | | | | | | |
| Brief Happiness | -0.04 | 0.08 | -0.04 | -0.20 | 0.11 | 0.582 |
| Brief Satisfaction | 0.08 | 0.17 | 0.04 | -0.12 | 0.20 | 0.619 |
| Positive Emotions | -0.16 | 0.12 | -0.10 | -0.25 | 0.06 | 0.202 |
| Negative Emotions | 0.03 | 0.13 | 0.02 | -0.14 | 0.18 | 0.815 |
| Life Satisfaction | 0.29 | 0.11 | 0.21 | 0.06 | 0.35 | 0.008 |
| Mindful Attention | 0.32 | 0.12 | 0.20 | 0.05 | 0.35 | 0.011 |
| Autonomy | 0.17 | 0.10 | 0.14 | -0.02 | 0.29 | 0.076 |
| Competence | 0.19 | 0.09 | 0.16 | 0.00 | 0.31 | 0.046 |
| Connectedness | -0.02 | 0.12 | -0.02 | -0.17 | 0.14 | 0.848 |
| Depression | -0.08 | 0.08 | -0.08 | -0.23 | 0.08 | 0.321 |
| Loneliness | -0.08 | 0.08 | -0.09 | -0.24 | 0.07 | 0.271 |
| Self-Esteem | 0.21 | 0.07 | 0.24 | 0.09 | 0.38 | 0.003 |
| Stress | -0.16 | 0.08 | -0.15 | -0.30 | 0.01 | 0.061 |
| Health | 2.14 | 2.00 | 0.09 | -0.07 | 0.24 | 0.286 |
| **Hypothesis 1. Digital Diet vs. Both Controls:** | | | | | | |
| Brief Happiness | 0.03 | 0.07 | 0.03 | -0.09 | 0.15 | 0.611 |
| Brief Satisfaction | 0.15 | 0.15 | 0.06 | -0.06 | 0.18 | 0.321 |
| Positive Emotions | -0.03 | 0.11 | -0.02 | -0.14 | 0.10 | 0.773 |
| Negative Emotions | -0.08 | 0.10 | -0.05 | -0.17 | 0.07 | 0.416 |
| Life Satisfaction | 0.33 | 0.09 | 0.21 | 0.10 | 0.32 | < .001 |
| Mindful Attention | 0.36 | 0.10 | 0.21 | 0.10 | 0.32 | < .001 |
| Autonomy | 0.23 | 0.08 | 0.17 | 0.05 | 0.28 | 0.006 |
| Competence | 0.21 | 0.08 | 0.15 | 0.04 | 0.27 | 0.011 |
| Connectedness | 0.05 | 0.10 | 0.03 | -0.09 | 0.15 | 0.665 |
| Depression | -0.09 | 0.07 | -0.08 | -0.2 | 0.04 | 0.196 |
| Loneliness | -0.15 | 0.07 | -0.13 | -0.25 | -0.01 | 0.030 |
| Self-Esteem | 0.26 | 0.06 | 0.25 | 0.14 | 0.36 | < .001 |
| Stress | -0.19 | 0.07 | -0.17 | -0.28 | -0.05 | 0.006 |

(*Continued*)

**Table 1.**  (Continued)

|  |  |  |  | Partial *r* 95% CI |  |  |
|---|---|---|---|---|---|---|
| Variable | *b* | *b* SE | Partial *r* | LL | UL | *p* |
| Health | 3.00 | 1.80 | 0.10 | -0.02 | 0.22 | 0.097 |

*Note*. Regressed change models. Hypothesized condition dummy codes predicting $T_2$ scores, controlling for $T_1$ scores. Positive *b*s/*r*s suggest the treatment group (Digital Diet) reported greater increases than the reference group (Water Diet, No Diet, or Both Controls). Negative *b*s/*r*s suggest the treatment group reported greater decreases than the reference group. CI = confidence interval; LL = lower limit; UL = upper limit.

### Exploratory: Restricting Digital media vs. Social media

We also compared the Digital Diet and Social Diet conditions directly in an exploratory manner (see Table 3). Given that participants in the Digital Diet condition experienced several benefits not shared by the Social Diet group, we coded the Social Diet as the reference group. Those in the Digital Diet condition reported greater increases in autonomy (partial *r* = .17, *p* = .045), self-esteem (partial *r* = .24, *p* = .004), and health (partial *r* = .17, *p* = .045) than those in the Social Diet group, as well as bigger decreases in negative emotions (partial *r* = −.17, *p* = .044). No significant differences between the two conditions emerged for brief happiness, brief satisfaction, positive emotions, life satisfaction, mindful attention, competence, connectedness, depression, loneliness, or stress.

### Discussion

In summary, Gen Z individuals who were asked to reduce their digital media or social media use for a little over a week appeared to carry out this charge relatively successfully. Notably, participants assigned to restrict their digital media use (i.e., time spent on their smartphones) experienced several benefits, including higher life satisfaction, mindfulness, autonomy, competence, and self-esteem, as well as reduced loneliness and stress. In contrast, those assigned to restrict their social media use (i.e., time spent on Facebook, Instagram, Twitter, and Snapchat) experienced relatively few benefits (increased life satisfaction and mindfulness) and even some costs, depending on the control group comparison (decreased positive emotions and increased negative emotions). Overall, the significant effects were small, but not tiny. Indeed, the effect sizes were often larger than those found in previous correlational research [14–17].

However, relative to controls, restricting digital media or social media did not improve emotional well-being or depression—critical mental health outcomes frequently debated in the correlational literature. Partial *r* effect sizes for those mental health outcomes were indeed close to zero, with the exception of positive and negative emotions for participants restricting social media, which reflected backfiring effects. Notably, our social media restriction findings are intriguingly consistent with Przybylski and colleagues' (2021) one-day social media fast study, which also found backfiring effects [48]. Further, our results are especially interesting considering recent baseline cross-sectional analyses we conducted with the same participants, which revealed social media use to be negatively associated with subjective well-being [r = −.16; 8]. In other words, when Gen Z individuals entered our study, the more social media they reported using, the less happy they were. However, prompting them to reduce their social media use did not make them any happier.

Our exploratory analyses comparing the two experimental conditions directly provided further evidence that reducing smartphone use was more beneficial than reducing social media use. Relative to those restricting social media time, participants who reduced their smartphone

**Table 2.** Hypothesis 2 Social diet comparisons.

| Variable | b | b SE | Partial r | Partial r 95% CI LL | UL | p |
|---|---|---|---|---|---|---|
| **Hypothesis 2. Social Diet vs. Water Diet:** | | | | | | |
| Brief Happiness | 0.14 | 0.08 | 0.14 | -0.01 | 0.27 | 0.066 |
| Brief Satisfaction | 0.19 | 0.16 | 0.08 | -0.06 | 0.23 | 0.258 |
| Positive Emotions | -0.06 | 0.14 | -0.03 | -0.18 | 0.11 | 0.648 |
| Negative Emotions | 0.10 | 0.11 | 0.07 | -0.08 | 0.21 | 0.365 |
| Life Satisfaction | 0.20 | 0.11 | 0.14 | -0.01 | 0.28 | 0.065 |
| Mindful Attention | 0.31 | 0.11 | 0.21 | 0.06 | 0.34 | 0.005 |
| Autonomy | 0.04 | 0.10 | 0.03 | -0.12 | 0.17 | 0.706 |
| Competence | 0.07 | 0.10 | 0.05 | -0.09 | 0.20 | 0.476 |
| Connectedness | 0.00 | 0.12 | 0.00 | -0.14 | 0.15 | 0.979 |
| Depression | 0.00 | 0.07 | 0.00 | -0.14 | 0.15 | 0.961 |
| Loneliness | -0.07 | 0.08 | -0.06 | -0.21 | 0.08 | 0.406 |
| Self-Esteem | 0.05 | 0.07 | 0.06 | -0.09 | 0.20 | 0.455 |
| Stress | -0.08 | 0.08 | -0.07 | -0.21 | 0.07 | 0.333 |
| Health | -0.31 | 2.23 | -0.01 | -0.16 | 0.14 | 0.891 |
| **Hypothesis 2. Social Diet vs. No Diet:** | | | | | | |
| Brief Happiness | 0.00 | 0.08 | 0.00 | -0.17 | 0.16 | 0.960 |
| Brief Satisfaction | 0.10 | 0.18 | 0.05 | -0.12 | 0.21 | 0.559 |
| Positive Emotions | -0.27 | 0.13 | -0.17 | -0.32 | -0.01 | 0.043 |
| Negative Emotions | 0.34 | 0.12 | 0.23 | 0.07 | 0.37 | 0.006 |
| Life Satisfaction | 0.14 | 0.12 | 0.09 | -0.07 | 0.25 | 0.264 |
| Mindful Attention | 0.21 | 0.13 | 0.13 | -0.03 | 0.29 | 0.109 |
| Autonomy | -0.03 | 0.10 | -0.03 | -0.19 | 0.14 | 0.737 |
| Competence | 0.05 | 0.10 | 0.04 | -0.13 | 0.20 | 0.664 |
| Connectedness | -0.09 | 0.12 | -0.06 | -0.22 | 0.11 | 0.485 |
| Depression | 0.03 | 0.08 | 0.03 | -0.13 | 0.19 | 0.730 |
| Loneliness | 0.03 | 0.09 | 0.03 | -0.13 | 0.19 | 0.701 |
| Self-Esteem | -0.01 | 0.08 | -0.01 | -0.18 | 0.15 | 0.871 |
| Stress | -0.01 | 0.09 | -0.01 | -0.17 | 0.15 | 0.900 |
| Health | -2.37 | 2.26 | -0.09 | -0.24 | 0.08 | 0.297 |
| **Hypothesis 2. Social Diet vs. Both Controls:** | | | | | | |
| Brief Happiness | 0.09 | 0.07 | 0.07 | -0.05 | 0.19 | 0.231 |
| Brief Satisfaction | 0.15 | 0.16 | 0.06 | -0.06 | 0.18 | 0.340 |
| Positive Emotions | -0.15 | 0.12 | -0.08 | -0.20 | 0.04 | 0.213 |
| Negative Emotions | 0.19 | 0.10 | 0.12 | -0.01 | 0.23 | 0.060 |
| Life Satisfaction | 0.18 | 0.10 | 0.11 | -0.01 | 0.23 | 0.079 |
| Mindful Attention | 0.26 | 0.10 | 0.16 | 0.04 | 0.27 | 0.012 |
| Autonomy | 0.01 | 0.09 | 0.01 | -0.12 | 0.13 | 0.932 |
| Competence | 0.06 | 0.09 | 0.04 | -0.08 | 0.16 | 0.507 |
| Connectedness | -0.03 | 0.11 | -0.02 | -0.14 | 0.11 | 0.805 |
| Depression | 0.01 | 0.07 | 0.01 | -0.11 | 0.13 | 0.858 |
| Loneliness | -0.03 | 0.07 | -0.02 | -0.14 | 0.10 | 0.703 |
| Self-Esteem | 0.03 | 0.07 | 0.02 | -0.10 | 0.15 | 0.691 |
| Stress | -0.05 | 0.07 | -0.04 | -0.16 | 0.08 | 0.486 |

(*Continued*)

**Table 2.** (*Continued*)

| Variable | b | b SE | Partial r | Partial r 95% CI LL | Partial r 95% CI UL | p |
|---|---|---|---|---|---|---|
| Health | -1.21 | 1.98 | -0.04 | -0.16 | 0.08 | 0.544 |

*Note.* Regressed change models. Hypothesized condition dummy codes predicting $T_2$ scores, controlling for $T_1$ scores. Positive $b$s/$r$s suggest the treatment group (Social Diet) reported greater increases than the reference group (Water Diet, No Diet, or Both Controls). Negative $b$s/$r$s suggest the treatment group reported greater decreases than the reference group. CI = confidence interval; LL = lower limit; UL = upper limit.

time reported greater increases in autonomy, self-esteem, and health, as well as greater decreases in negative emotions.

Overall, our study is one of a small handful of recent large, controlled experimental studies investigating the question of what happens when people deliberately restrict their screen time. Accordingly, it makes several important contributions to the existing literature on digital media use and well-being. Unlike previous research that has focused on restriction during shorter time frames (e.g., 1 day) or specific circumstances (e.g., at dinner), our experimental design implemented a general restriction over an extended period (8–10 days). Furthermore, we utilized objective measures of smartphone use (coded iPhone Screen Time screenshots), providing more reliable time use data than self-reported measures. Our study also restricted a wide range of social media apps, including Facebook, Instagram, Twitter, and Snapchat, offering a more comprehensive analysis of the effects of restricting social media. Additionally, we compared the outcomes of smartphone versus social media restriction, and assessed a number of outcomes not previously examined in prior studies. Lastly, we implemented a stronger neutral activity control group (water restriction) to better control for experimental demand effects. These methodological advancements address significant gaps in the literature and provide a

**Table 3.** Exploratory Digital diet vs. Social diet comparisons.

| Variable | b | b SE | Partial r | Partial r 95% CI LL | Partial r 95% CI UL | p |
|---|---|---|---|---|---|---|
| **Exploratory. Digital Diet vs. Social Diet:** | | | | | | |
| Brief Happiness | -0.03 | 0.08 | -0.03 | -0.19 | 0.14 | 0.728 |
| Brief Satisfaction | -0.03 | 0.16 | -0.02 | -0.18 | 0.15 | 0.834 |
| Positive Emotions | 0.12 | 0.14 | 0.07 | -0.10 | 0.23 | 0.399 |
| Negative Emotions | -0.27 | 0.13 | -0.17 | -0.32 | 0.00 | 0.044 |
| Life Satisfaction | 0.16 | 0.11 | 0.12 | -0.04 | 0.28 | 0.139 |
| Mindful Attention | 0.08 | 0.13 | 0.05 | -0.11 | 0.22 | 0.531 |
| Autonomy | 0.21 | 0.10 | 0.17 | 0.00 | 0.32 | 0.045 |
| Competence | 0.14 | 0.10 | 0.11 | -0.05 | 0.27 | 0.180 |
| Connectedness | 0.06 | 0.12 | 0.04 | -0.12 | 0.21 | 0.613 |
| Depression | -0.11 | 0.07 | -0.12 | -0.28 | 0.04 | 0.140 |
| Loneliness | -0.12 | 0.08 | -0.12 | -0.27 | 0.05 | 0.168 |
| Self-Esteem | 0.23 | 0.08 | 0.24 | 0.08 | 0.38 | 0.004 |
| Stress | -0.16 | 0.10 | -0.14 | -0.29 | 0.03 | 0.100 |
| Health | 4.55 | 2.25 | 0.17 | 0.00 | 0.32 | 0.045 |

*Note.* Regressed change models. Hypothesized condition dummy codes predicting $T_2$ scores, controlling for $T_1$ scores. Positive $b$s/$r$s suggest the treatment group (Digital Diet) reported greater increases than the reference group (Social Diet). Negative $b$s/$r$s suggest the treatment group reported greater decreases than the reference group. CI = confidence interval; LL = lower limit; UL = upper limit.

more nuanced understanding of how digital media use influences well-being. Interestingly, our experimental findings dovetail with the nuanced and mixed results from correlational work. However, much more research is needed, including replications and mechanism studies with even larger sample sizes, but accumulating evidence suggests that it may not be the amount of time spent on digital media or social media that meaningfully impacts well-being, but how, why, when, and where one spends that time [66–68].

Although reducing time spent on digital media did have some meaningful positive effects, the present investigation does not lead to the conclusion that smartphones and social media are especially harmful to young people. However, revelations from the "Facebook Files"—a collection of stories published in 2021 at the *Wall Street Journal* based on hundreds of pages of leaked documents from inside Facebook—demonstrate that social media platforms are not always optimized for user well-being [69]. For example, internal company documents showed that teens who struggle with mental health report that Instagram makes it worse. But not all social networking sites elicit identical effects. Another study assessing baseline cross-sectional findings with the same participants in this study showed that Facebook use is associated with lower well-being, while Snapchat use is associated with higher well-being [8]. In other words, smartphone apps can be designed to increase (or decrease) user happiness.

Thus, extreme vilification of these technologies may be unwarranted. Smartphone apps may simply represent tools that can be tailored for better or worse outcomes. Historically, new inventions (e.g., novels, radios) have often prompted socially contagious technology panics [70]. For example, in 1680, philosopher and mathematician Gottfried Wilhem von Leibniz questioned the usefulness of the printing press, suggesting that "the horrible mass of books that keeps growing might lead to a fall back into barbarism" [71]. Yet despite early concerns, many new technologies are ultimately adopted into daily life, and go on to provide myriad benefits to humanity, such as improved health and wealth [72].

## Limitations and future directions

This study is subject to several limitations that may seed future work. Because our aim was to examine the effects of restricting digital and social media on American Gen Z individuals (mostly ages 18–25), our findings may not generalize to other populations. Future experimental restriction studies should recruit adolescent minors (younger than 18), as well as older adults (ages 26 and older) from other generations (e.g., Millennials, Gen Xers, Boomers), to observe effects on people from different age groups and cohorts. For example, younger people might benefit more than older people from restricting social media because they may be more susceptible to social influence or are heavier users [41,73]. To increase generalizability, such studies also need to be conducted in different cultures (e.g., individualist vs. collectivist, tight vs. loose), settings (e.g., urban vs. rural), and languages, as such variables could moderate effects. For example, vertical collectivism (i.e., positioning oneself hierarchically within an ingroup) is positively associated with nomophobia (fear of being without one's smartphone) [74]. Thus, members of collectivist cultures may experience backfiring effects when restricting their smartphone use.

Furthermore, although relatively large for a high effort "shortitudinal" intervention [18], our sample size ($N \geq 250$) may still not have been large enough. After all, the correlational research that inspired this work relied on exceptionally large sample sizes with Ns up to over a million participants [14–17]. As statistical significance depends on both effect size and sample size, our sample ($N = 338$) may have been too small to significantly detect the very small effects generally observed in the correlational literature. Of course, even if those small effects had been detected, the debate about whether such effects are practically meaningful would likely

continue. Regardless, researchers might consider using big data, Many Labs, and/or Psychological Science Accelerator approaches in the future to achieve even greater generalizability and statistical power, as well as to obtain more robust effect size estimates [75–77].

Although the participants in our study did successfully restrict the amount of time they spent on their smartphones and social media apps, they did not restrict as much as we would have wished. For example, participants' Screen Time screenshots revealed that the Digital Diet group restricted their iPhone use by an average of 115 minutes per day. However, they were still using their iPhones for about 211 minutes per day at posttest. Similarly, the Social Diet group restricted their social media use by 68 minutes per day, but they were still using social media for an average of 50 minutes per day at posttest. If participants had reduced their digital and social media use to negligible levels, the effects may have been stronger or reversed. Future researchers may identify new ways to induce individuals to reduce their screen time more effectively. Some studies have also tried paying users to deactivate specific social media accounts [40,78], but this requires recruiting self-selected participants willing to reduce their use. It is important to note that our participants did not self-select into the intervention. Thus, individuals in natural settings who wish to limit their screen time and actively take steps to do so may also experience benefits.

To be sure, future studies should experimentally restrict digital media for longer periods of time (e.g., 1, 3, or 6 months) [40,47]. Some writers and researchers have argued that smartphones and social media are addictive, and designed to hopelessly hook users [79–81]. By this reasoning, it is possible that participants who restricted their social media use may have reported greater levels of negative emotions (e.g., anger, sadness) than controls because they were experiencing something akin to withdrawal symptoms. If these participants had restricted social media for longer periods of time, they may have started to replace it with non-screen habits like exercising and gathering with friends, and thus may have ultimately enjoyed greater benefits. Alternatively, if the complementarity hypothesis holds true—that smartphones enhance well-being by offering a variety of stimulating information and activities—then longer-term restrictions of social media may exacerbate negative effects by limiting access to enjoyable experiences.

Past research spotlights that smartphones can be helpful or harmful to well-being, depending on the situation [22,23]. Likely content and context matter more than the amount of time spent [66]. Future studies could explore these issues more deeply so that investigators can determine how, why, when, and where to best use smartphones and social media to optimize well-being and related constructs. Device manufacturing, operating system, and app design companies would likely benefit from such research, allowing them to remodel their technologies to better support user well-being. Happier customers likely translate into higher sales and better engagement, thereby feasibly facilitating an improved digital environment for all.

Finally, our findings hold potential implications for policymakers, educators, and mental health professionals dealing with Gen Z. Policymakers may consider the benefits of reduced smartphone use over mere social media restriction when formulating guidelines or regulations related to digital media consumption. Specific policy measures could include setting screen time limits for minors and encouraging schools to incorporate balanced digital practices. Additionally, workplace policies might be designed to include breaks from digital devices to increase productivity and reduce digital fatigue. Educators could encourage healthy digital habits, by incorporating digital literacy programs into their curricula that teach students how to use these tools most efficiently. Examples of digital literacy modules could include practical exercises on mindful technology use and projects that help students understand the impacts of their digital habits on mental health. Educational technology specialists could develop these programs to ensure they balance technology use with educational outcomes. Mental health

professionals could use these insights to guide therapeutic strategies, focusing on the benefits of intentional and mindful smartphone use. Specific therapeutic techniques might include digital detox interventions or mindfulness practices aimed at reducing screen time. Family therapists could also counsel parents on managing their children's digital media usage to foster healthier digital habits within families.

In sum, our results advocate for a nuanced approach to digital media, highlighting the importance of not just limiting screen time but promoting healthier, more mindful interactions with technology. Future research could explore longitudinal methodologies that track media use changes over time, experimental studies to test the effectiveness of specific interventions, and cross-cultural research to examine how these findings apply in different societal contexts.

## Conclusion

Most people want to be happy [82], and many report being happier with a smartphone than without one [83]. Perhaps this is why smartphones have spread faster than any technology in human history [20], prompting anxiety about their effects on human happiness and health. Is there value and virtue in reducing individuals' reliance on screens? Our results show that restricting smartphone use for a week does appear to grant some benefits (e.g., greater life satisfaction, decreased loneliness), but reducing social media use (not just Facebook, but also Instagram, Snapchat, Twitter, etc.) appears to provide few advantages (greater mindfulness) and perhaps even some costs (more negative emotion). Overall, the effect sizes were generally larger than what has been reported in previous correlational research, yet still relatively small. Overall, these findings highlight the need for informed, evidence-based strategies to navigate the complex relationship between digital media use and mental health.

## Supporting information

**S1 File. Supplemental materials.**
(DOCX)

## Acknowledgments

We would like to thank the following people for their useful comments, suggestions, and contributions to this project: Steve Cole, William L. Dunlop, Megan M. Fritz, Sharath Chandra Guntuku, Seth Margolis, Chris C. Martin, Kate Sweeny, Jean Twenge, Lyle Ungar, and David A. Walsh. We would also like to thank Gabrielle Celaya, our undergraduate research coordinator, as well as our team of undergraduate research assistants, who dedicated innumerable hours to collecting these data and coding screenshots: Gabriela Alaniz, Estefanie Altuna, Fiorella Atoche Fernandez, Alana Barr, Joshua Bateman, S. Gokce Boz, Tia Calhoun, Ritz Cardona, Elizabeth Castellanos, Desire Cervantes, Elizabeth Dawson, Tia Elpusan, Arlene Enriquez-Rizo, Alvin Fong, Alec Frias, Monica Guirgus, Alan Guo, Victor Hill Jr., Margaret Lobermann, Taraji Long, Mary Mansour, Ashley Marin, Juliana Mena, Soumil Nariani, Vi Nguyen, Chuka Okolie, Shreya Parthiban, Yocelyn Ramirez-Pena, Jasmine Ramos, Samantha Rodriguez, Julie Salama, Mei Takeda, Samantha Thomas, and Christin Timmons.

## Author Contributions

**Conceptualization:** Lisa C. Walsh, Sonja Lyubomirsky.

**Data curation:** Lisa C. Walsh, Annie Regan, Karynna Okabe-Miyamoto, Sonja Lyubomirsky.

**Formal analysis:** Lisa C. Walsh.

**Investigation:** Lisa C. Walsh, Annie Regan, Karynna Okabe-Miyamoto, Sonja Lyubomirsky.

**Methodology:** Lisa C. Walsh, Annie Regan, Karynna Okabe-Miyamoto, Sonja Lyubomirsky.

**Project administration:** Lisa C. Walsh, Annie Regan, Karynna Okabe-Miyamoto, Sonja Lyubomirsky.

**Supervision:** Sonja Lyubomirsky.

**Writing – original draft:** Lisa C. Walsh, Sonja Lyubomirsky.

**Writing – review & editing:** Lisa C. Walsh, Annie Regan, Karynna Okabe-Miyamoto, Sonja Lyubomirsky.

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
