## [Decision Letter · Decision Letter 0]

24 Apr 2023

PONE-D-23-01416Does Putting Down Your Smartphone Make You Happier? The Effects of Restricting Digital Media on Well-BeingPLOS ONE

Dear Dr. Walsh,

Thank you for submitting your manuscript to PLOS ONE. After careful consideration, we feel that it has merit but does not fully meet PLOS ONE’s publication criteria as it currently stands. Therefore, we invite you to submit a revised version of the manuscript that addresses the points raised during the review process.

We look forward to receiving your revised manuscript.

Kind regards,

Alastair Van Heerden

Academic Editor

PLOS ONE

2. Please ensure that you include a title page within your main document. You should list all authors and all affiliations as per our author instructions and clearly indicate the corresponding author.

Reviewers' comments:

Reviewer's Responses to Questions

**Comments to the Author**

1. Is the manuscript technically sound, and do the data support the conclusions?

Reviewer #1: Yes

Reviewer #2: Yes

2. Has the statistical analysis been performed appropriately and rigorously? 

Reviewer #1: Yes

Reviewer #2: Yes

3. Have the authors made all data underlying the findings in their manuscript fully available?

Reviewer #1: Yes

Reviewer #2: Yes

4. Is the manuscript presented in an intelligible fashion and written in standard English?

Reviewer #1: Yes

Reviewer #2: Yes

5. Review Comments to the Author

Reviewer #1: Congratulations! The study is relevant, methodologically quite sound, and provides novel and interesting findings presented in a clear and straightforward way and interpreted correctly. I particularly enjoyed reading the literature review - it is very informative and well-written. The discussion is relevant and to the point.

Overall, the paper is so good that I do not have any suggestions for improvement: if it's not broken, don't fix it. Thank you for conducting this study and I'm looking forward to seeing it published.

Reviewer #2: Overall Structure: The manuscript is well-structured, starting with an introduction to the ubiquity of smartphones and digital media, followed by a review of existing literature on the impact of digital media and social media on well-being. The paper then presents the need for further research and the objective of the study.

Clarity: The manuscript is generally clear, but there is room for improvement in certain areas. The introduction section could benefit from a sharper focus on the relationship between smartphone use and well-being. Additionally, it may be helpful to provide more context on the specific research questions and hypotheses in the introductory section.

Methodology: The methodology is well-described, with information on participant selection, study design, and measures used. The study employs a between-subjects design with four conditions: Digital Diet, Social Diet, Water Diet (active control), and No Diet (measurement-only control). The study focuses on Gen Z individuals and uses a range of well-established measures to assess well-being and related constructs.

Results

1. The authors should provide more information on the randomization process, including how participants were allocated to different groups and any stratification factors used to ensure balance across groups.

2. The sample size should be justified based on a power analysis or other relevant calculations, to ensure that the study has sufficient power to detect meaningful differences between groups.

3. The authors should consider conducting a sensitivity analysis to assess the robustness of their findings to potential biases, such as attrition or nonresponse.

Other suggestions for improvement

1. The authors should address the potential for selection bias in their sample, given that the participants were recruited from a specific population (Gen Z individuals, mainly ages 18-25). They should discuss how this might limit the generalizability of their findings and consider conducting sensitivity analyses or subgroup analyses to account for potential biases.

2. The authors could consider including a mediation or moderation analysis to investigate the potential mechanisms underlying the observed effects. This would help to provide deeper insights into how and why restricting digital media and social media use might influence psychological outcomes.

3. In the discussion, the authors should consider the potential implications of their findings for policymakers, educators, and mental health professionals. For example, they could discuss how their results might inform strategies for promoting healthy digital media and social media use among young people and address the potential for digital interventions to improve psychological well-being.

4. In the sentence "For ease of interpretation, difference scores are presented in raw (not log-transformed) form.", the term "raw" should be replaced with "untransformed" for clarity and precision.

5. In the sentence "However, as recent revelations from The Wall Street Journal’s (2021) “Facebook Files” demonstrate, social media platforms are not always optimized for user well-being [61].", consider adding an explanation or brief summary of the "Facebook Files" for readers who may be unfamiliar with the issue.

6. In the sentence "Of course, even if those small effect sizes are made detectable, the debate about whether such effects matter will likely continue.", consider rephrasing to "Of course, even if those small effect sizes are detected, the debate about whether such effects are practically significant will likely continue."

7. In the sentence "Smartphone apps may merely represent tools that can be tailored for better or worse.", consider rephrasing as "Smartphone apps may simply represent tools that can be tailored for better or worse outcomes."

6. PLOS authors have the option to publish the peer review history of their article (what does this mean?). If published, this will include your full peer review and any attached files.

Reviewer #1: No

Reviewer #2: No

---

## [Author Response · Author response to Decision Letter 0]

6 Jul 2023

Please see attached response to reviewers.

---

## [Decision Letter · Decision Letter 1]

16 Nov 2023

PONE-D-23-01416R1Does Putting Down Your Smartphone Make You Happier? The Effects of Restricting Digital Media on Well-BeingPLOS ONE

Dear Dr. Walsh,

Thank you for submitting your manuscript to PLOS ONE. After careful consideration, we feel that it has merit but does not fully meet PLOS ONE’s publication criteria as it currently stands. Therefore, we invite you to submit a revised version of the manuscript that addresses the points raised during the review process.

We look forward to receiving your revised manuscript.

Kind regards,

Ali B. Mahmoud, Ph.D.

Academic Editor

PLOS ONE

Journal Requirements:

Reviewers' comments:

Reviewer's Responses to Questions

**Comments to the Author**

1. If the authors have adequately addressed your comments raised in a previous round of review and you feel that this manuscript is now acceptable for publication, you may indicate that here to bypass the “Comments to the Author” section, enter your conflict of interest statement in the “Confidential to Editor” section, and submit your "Accept" recommendation.

Reviewer #3: (No Response)

2. Is the manuscript technically sound, and do the data support the conclusions?

Reviewer #3: Yes

3. Has the statistical analysis been performed appropriately and rigorously? 

Reviewer #3: Yes

4. Have the authors made all data underlying the findings in their manuscript fully available?

Reviewer #3: Yes

5. Is the manuscript presented in an intelligible fashion and written in standard English?

Reviewer #3: Yes

6. Review Comments to the Author

Reviewer #3: This document addresses a relevant topic about the study of happiness and the value that the use of digital media, social media, has on it. To do this, a comparison is made with the restrictive use of these two concepts and the use applied in some water applications in contrast to not having any restrictions. This in order to observe the differences and appreciate what effects they have on the perception of a state of well-being.

The authors make a methodological development very appropriate to the type of study they propose and apply strategies that are made up of statistical methods appropriate for the type of study.

They present results in accordance with the initial approaches and an exploratory approach is manifested for the type of study that aims to find the effects of the study in its analysis and in the relationships found between the constructs that make up each of the variables.

In its results, once the statistical studies are presented, a methodological and statistical rigor is evident that is very appropriate for the type of scientific product that is proposed for this journal. I consider that the paper meets an adequate quality to be considered for publication.

I recommend the following adjustments in its approach and development:

1. Establish a table of authors at the end of each of the sections where the study variables are defined. In this case, at the end of digital media, social media, water restrictions, and where appropriate, support for the scenario where there is no type of restriction.

In this table, I recommend including the most relevant previous documents that support the theoretical model on which the proposed model is based.

2. Add a graphic diagram to illustrate the development of the methodological diagram developed during this document. To do this, add in each phase an explanation about the inputs from the previous phase and what elements or values are delivered in the next step, as a process diagram.

3. In your conclusions, delve into the effects that develop from the findings in your study, what elements are added to the literature, and what gaps are closed with the results of this research, likewise, in case they are projected new forms of questions, new methods or elements that propose improvements in the work proposals, include them in your explanation of future work and confirmation of obtained results.

4. Expand the explanation of the lines of research that open from this research, as well as the elements that are required for these results to be extrapolated or generalized in public policies or in new conditions that promote well-being

I consider that these are minor changes, in the theoretical, methodological, and technical aspects, it seems to me that the rigor is of a very well-founded level.

Analyzing the impacts of restrictions on digital media, social media, water usage, and a scenario with no restrictions can indeed contribute to our overall happiness. Restrictions on digital media and social media can curb information overload, enhancing mental well-being. Water usage restrictions foster sustainability which supports a healthier environment and future generations' happiness. However, a complete absence of restrictions may lead to detrimental consequences in both digital and environmental domains, affecting our overall contentment. Hence, a balanced approach that combines responsible usage with thoughtful regulation can promote happiness by safeguarding our well-being and the planet's health

7. PLOS authors have the option to publish the peer review history of their article (what does this mean?). If published, this will include your full peer review and any attached files.

Reviewer #3: **Yes: **Eduardo Ahumada-Tello

---

## [Author Response · Author response to Decision Letter 1]

18 May 2024

Please see the attached Response to Reviewers letter.

---

## [Decision Letter · Decision Letter 2]

26 Jun 2024

Does Putting Down Your Smartphone Make You Happier? The Effects of Restricting Digital Media on Well-Being

PONE-D-23-01416R2

Dear Dr. Walsh,

We’re pleased to inform you that your manuscript has been judged scientifically suitable for publication and will be formally accepted for publication once it meets all outstanding technical requirements.

Kind regards,

Ali B. Mahmoud, Ph.D.

Academic Editor

PLOS ONE

Additional Editor Comments (optional):

Reviewers' comments:

Reviewer's Responses to Questions

**Comments to the Author**

1. If the authors have adequately addressed your comments raised in a previous round of review and you feel that this manuscript is now acceptable for publication, you may indicate that here to bypass the “Comments to the Author” section, enter your conflict of interest statement in the “Confidential to Editor” section, and submit your "Accept" recommendation.

Reviewer #3: All comments have been addressed

2. Is the manuscript technically sound, and do the data support the conclusions?

Reviewer #3: Yes

3. Has the statistical analysis been performed appropriately and rigorously? 

Reviewer #3: Yes

4. Have the authors made all data underlying the findings in their manuscript fully available?

Reviewer #3: Yes

5. Is the manuscript presented in an intelligible fashion and written in standard English?

Reviewer #3: Yes

6. Review Comments to the Author

Reviewer #3: The recommended changes to the document have been accepted and implemented. Tables of authors with relevant documents have been added as the other comments and recommendations made to authors:

Comments recommended in previous version:

1. Establish a table of authors at the end of each of the sections where the study variables are defined. In this case, at the end of digital media, social media, water restrictions, and where appropriate, support for the scenario where there is no type of restriction.

In this table, I recommend including the most relevant previous documents that support the theoretical model on which the proposed model is based.

2. Add a graphic diagram to illustrate the development of the methodological diagram developed during this document. To do this, add in each phase an explanation about the inputs from the previous phase and what elements or values are delivered in the next step, as a process diagram.

3. In your conclusions, delve into the effects that develop from the findings in your study, what elements are added to the literature, and what gaps are closed with the results of this research, likewise, in case they are projected new forms of questions, new methods or elements that propose improvements in the work proposals, include them in your explanation of future work and confirmation of obtained results.

4. Expand the explanation of the lines of research that open from this research, as well as the elements that are required for these results to be extrapolated or generalized in public policies or in new conditions that promote well-being

These modifications enhance the theoretical, methodological, and technical rigor, ensuring the paper meets publication standards.

I recommend that the authors confirmed the format and extension of the paper according to the journal regulations.

7. PLOS authors have the option to publish the peer review history of their article (what does this mean?). If published, this will include your full peer review and any attached files.

Reviewer #3: **Yes: **Eduardo Ahumada-Tello

---

## [Editor Report · Acceptance letter]

15 Jul 2024

PONE-D-23-01416R2 

PLOS ONE

Dear Dr. Walsh, 

I'm pleased to inform you that your manuscript has been deemed suitable for publication in PLOS ONE. Congratulations! Your manuscript is now being handed over to our production team.

Kind regards, 

on behalf of

Dr. Ali B. Mahmoud 

Academic Editor

PLOS ONE